# Competing risk of mortality in association studies of non-fatal events

**Petra Buzkova** *

Department of Biostatistics, University of Washington, Seattle, WA, United States of America

* buzkova@uw.edu

## Abstract

In geriatric research of non-fatal events, participants often die during the study follow-up without having the non-fatal event of interest. Cause-specific (CS) hazard regression and Fine-Gray (FG) subdistribution hazard regression are the two most common estimation approaches addressing such competing risk. We explain how the conventional CS approach and the FG approach differ and why many FG estimates of associations are counter-intuitive. Additionally, we clarify the indirect link between models for hazard and models for cumulative incidence. The methodologies are contrasted on data from the Cardiovascular Health Study, a population-based study in adults aged 65 years and older.

## Introduction

Many studies use the time to a non-fatal event as their primary outcome. In geriatric research, mortality often precludes individuals from reaching the end of the study and thus possibly prevents primary events from happening. A competing risk event is, by definition, an event that either hinders the observation of the event of interest or modifies the chance that this event occurs. Therefore, death is indeed a competing risk event.

FG subdistribution hazard regression is often the recommended methodology for competing risk scenarios. However, substantial confusion exists about the interpretation of the FG estimates [1–3]. A systematic review of the use and interpretation of the FG methods, conducted from medical literature in 2015 [2] and coauthored by one of the originators of the FG methodology, found that 91% of papers using FG methodology interpreted its estimates unclearly or incorrectly. This is an alarming state of practice, since incorrect interpretation of estimates may trigger incorrect conclusions and clinical decisions. This review excluded methodologically oriented publications, but even there the FG methodology is misrepresented. For instance, the authors of a paper guiding the use of competing risk methods for death in a gerontology journal incorrectly claim that the FG methodology "adjusts" for the associated risk of competing events [4]. The misuse of the FG methodology may be further encouraged by the fact that implementing FG estimators is extremely easy. There are many papers with high citation counts serving as a guide for clinicians for obtaining the FG estimates using statistical software where functions were build-in for convenience [5–7]. Some of these guidelines

**Data Availability Statement:** All files are from the Cardiovascular Health Study; https://chs-nhlbi.org/ Data repository is at biolincc.nhlbi.nih.gov/studies/chs/ available to researchers for requests.

**Funding:** This research was supported by contracts HHSN 268201200036C, HHSN

268200800007C, HHSN 268201800001C, N01HC 55222, N01HC 85079, N01HC 85080, N01HC 85081, N01HC 85082, N01HC 85083, N01HC 85086, and grants U01HL 080295 and U01HL 130114 from the National Heart, Lung, and Blood Institute (NHLBI), with additional contribution from the National Institute of Neurological Disorders and Stroke (NINDS). Additional support was provided by R01AG 023629 from the National Institute on Aging (NIA). A full list of principal CHS investigators and institutions can be found at CHS-NHLBI.org. The content is solely the responsibility of the authors and does not necessarily represent the official views of the National Institutes of Health. The funders had no role in study design, data collection and analysis, decision to publish, or preparation of the manuscript.

**Competing interests:** The authors have declared that no competing interests exist.

mislead researchers with incorrect interpretations of the estimates, e.g, some authors call FG subdistribution hazard ratios relative risk or incidence rate ratios [6].

The Cardiovascular Health Study (CHS) is ideal for studying the impact of the competing risk of death because 87% of the 5265 participants from the baseline 1992/93 visit died prior to the end of the clinical follow-up of 22 years. Furthermore, 87% of those deaths occurred in participants prior to an incident hip fracture, the event of our interest. A pie chart of the status of the CHS population at the end of the follow-up in Fig 1 shows that while only 13% experienced the primary event of incident hip fracture, 76% suffered from the competing event of death without a prior hip fracture.

Table 1, borrowed from a clinical paper [8], further summarized in Fig 2, lists the hazard ratio estimates that were obtained applying the traditional cause-specific Cox regression that censors individuals at death and the FG subdistribution hazard regression [9] in the CHS population. The methods sometimes yielded similar strengths of association, sometimes directionally concordant but quantitatively different strengths of association, and at other times estimates discordant in direction, with the FG subdistribution hazard regression suggesting effects in the opposite direction of well-understood and widely accepted associations [8].

In this article we study in detail the difference between the cause specific and subdistribution hazard definitions. We explain why we can observe such striking differences between the hazard ratio estimates, and discuss whether we can foresee how the estimates change when we switch the methodologies. We then focus on incidence rates, which are often overlooked. Incidence rates are simple to grasp, and the cause-specific and subdistribution approaches to these statistics clearly demonstrate the conceptual difference between the methodologies. Because it is sometimes suggested that the FG approach estimates cumulative incidence, we will clarify the link between hazard and cumulative incidence. The CHS is uniquely suited for demonstrating differences in these approaches owing to the long follow-up of several decades and the large proportion of now deceased participants.

## Hazard

In time to event data, the outcome of interest is not only whether or not an event occurred, but also the time during which an individual was at risk for the event. Hazard is based on fully utilizing time-to-event data. It is defined as the limit of the probability rate of having an event at time $t$ conditional on being at risk at that time. Denote $T$ the time to event, also called failure time, and $D$ the event type. Let us denote the primary event $k$.

The cause specific hazard at time $t$ for event $k$ is defined as

$$\lambda_k^{CS}(t) = \lim_{\Delta t \to 0} \frac{P(t \leq T < t + \Delta t, D = k | T \geq t)}{\Delta t}.$$

Those who have not yet experienced the event of interest or the competing event are at risk at time $t$. This approach is often called conditional since it conditions on not having had any type of event; $(|T \geq t)$. If an individual has a competing event, they are censored at that time and removed from the risk set from then on. In our scenario, cause specific hazard for incident hip fracture at time $t$ is defined in the population of alive and hip fracture-free individuals at time $t$.

The subdistribution (SD) hazard at time $t$ for event $k$, on the other hand, is defined as

$$\lambda_k^{SD}(t) = \lim_{\Delta t \to 0} \frac{P(t \leq T < t + \Delta t, D = k | T \geq t \cup (T < t, D \neq k)}{\Delta t}.$$

# Event Status

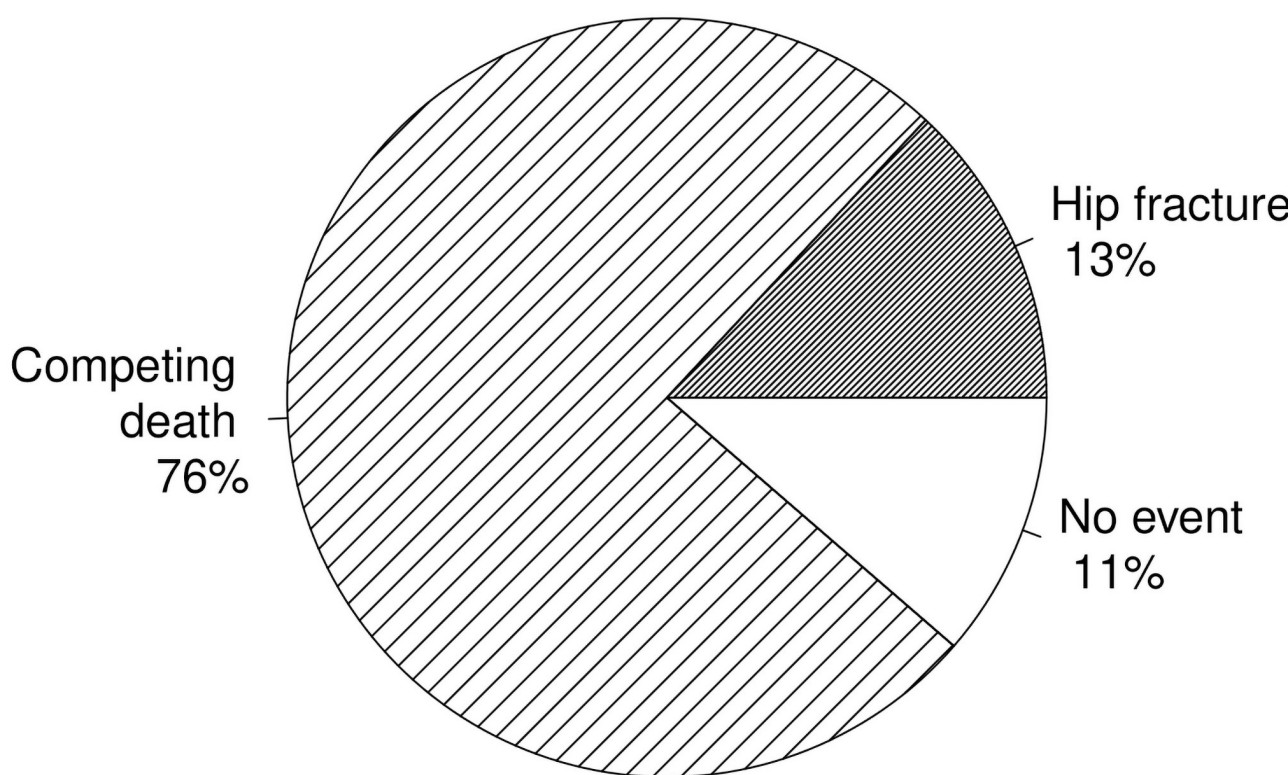

**Fig 1. Events status.** The 1992/93 CHS population status at the end of follow-up for incident hip fracture on June 30, 2014. 76% of the study participants suffered from the competing event of death.

**Table 1. Hazard ratio estimates for incident hip fracture, with mortality a competing risk.**

|  | Cause-specific hazard | | | FG subdistribution hazard | | |
|---|---|---|---|---|---|---|
|  | **HR** | **95%CI** | **p-val** | **HR** | **95%CI** | **p-val** |
| M1: |  |  |  |  |  |  |
| 5 years of age | 1.74 | (1.61, 1.87) | <0.01 | 1.16 | (1.09, 1.24) | <0.01 |
| male | 0.62 | (0.53, 0.74) | <0.01 | 0.49 | (0.41, 0.58) | <0.01 |
| black | 0.39 | (0.29, 0.52) | <0.01 | 0.38 | (0.29, 0.51) | <0.01 |
| M2s = M1+: |  |  |  |  |  |  |
| current smoking | 1.66 | (1.28, 2.14) | <0.01 | 1.17 | (0.90, 1.51) | 0.24 |
| diabetes | 1.19 | (0.94, 1.52) | 0.15 | 0.79 | (0.62, 1.01) | 0.06 |
| cystatin C eGFR* | 0.95 | (0.89, 1.02) | 0.19 | 1.09 | (1.02, 1.17) | <0.01 |

Model M1 is adjusting for demographic factors of age, gender and race. Models M2 are adjusting for the demographic factors and an additional risk factor of smoking, diabetes, and cystatin C-based estimated glomerular filtration rate (eGFR), in separate models.

* per15 ml/min/1.73 $m^2$

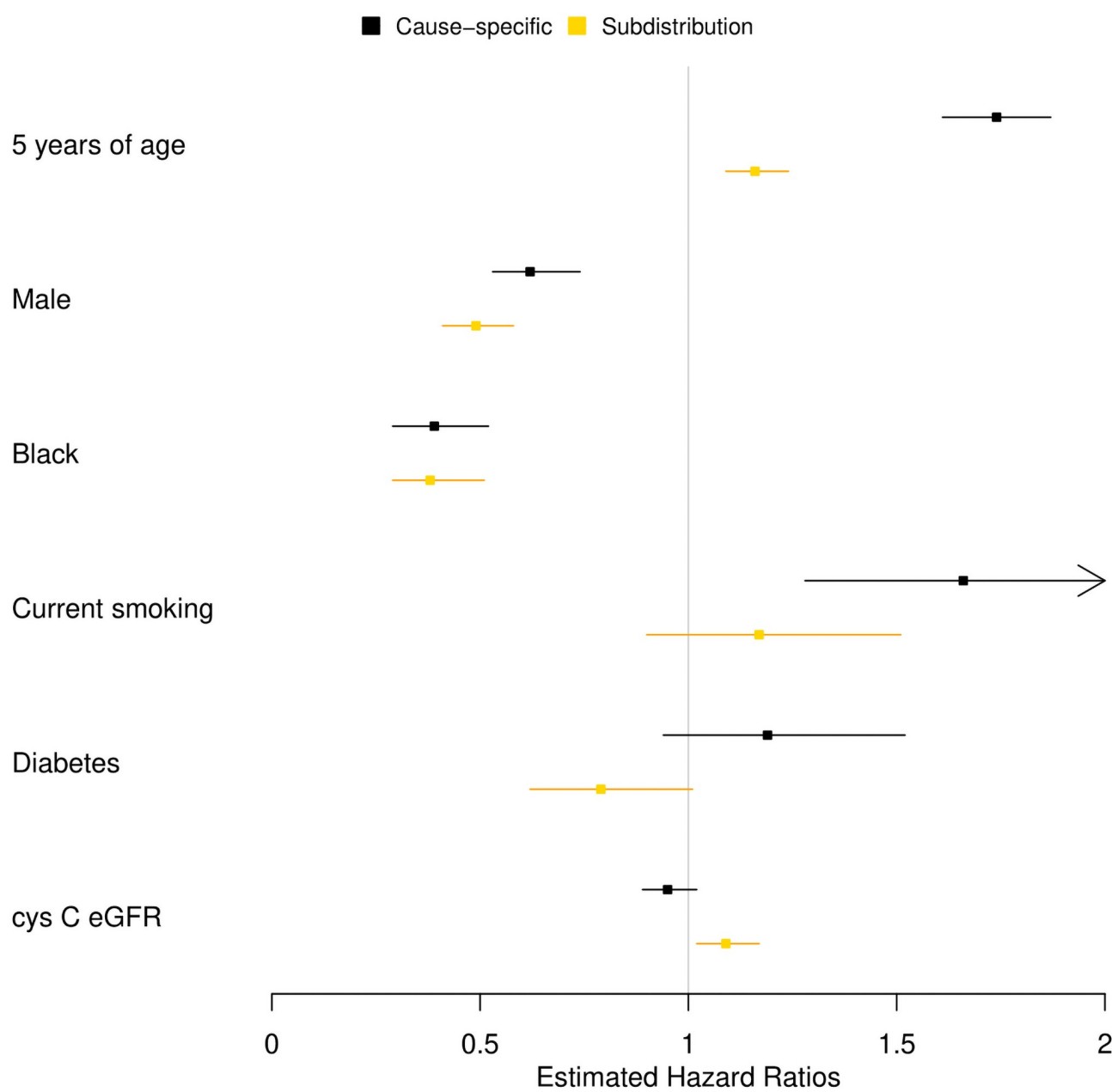

**Fig 2. Hazard ratios estimates.** Contrasting the magnitudes of the hazard ratio estimates for incident hip fracture between the cause-specific and FG subdistribution approaches.

The two definitions differ in the risk sets, that is, in what the probabilities of having an event $k$ at time $t$ condition on. In the subdistribution approach the risk set consists of those who have not yet had any type of event ($T \geq t$), and ($\cup$), additionally, of those who had a competing event that preceded time $t$ ($T < t$, $D \neq k$) [10]. This risk set is unnatural, consisting of these two very distinct groups of individuals [9]. In our scenario those groups are a population of alive hip fracture-free individuals and hip fracture-free individuals previously deceased. Thus, at risk are all individuals who have not experienced the event of interest, without regard to the competing event. This approach is called marginal, in contrast to the conditional cause-

specific approach. The term "subdistribution", defining the FG methodology, refers to the distribution of an improper random time variable whose density has an augmented mass at infinity due to competing events [9]. Having a competing event protects an individual indefinitely from experiencing the primary event, in spite of the fact that such an individual is retained in the risk set as if at risk for the primary event.

Both cause-specific and subdistribution hazard approaches typically model their hazard function $\lambda(t)$ using the same semiparametric Cox model [11]

$$\lambda(t) = \lambda_0(t) \exp\{\beta^T Z\},$$

where $Z$ is a vector of risk factors and $\lambda_0$ is an unspecified baseline hazard function under $Z = 0$. The parameters of interest are the $\beta$s, the log-hazard ratios.

The partial likelihood function for estimating the cause-specific hazard is

$$L(\beta) = \prod_{i=1}^{n} \left( \frac{\exp\{\beta^T Z_i\}}{\sum_{j \in R_i} \exp\{\beta^T Z_j\}} \right)^{I(D_i=k)}, \tag{1}$$

where we contrast an individual $i$ with an event $k$ to those in the risk set $R_i$, that is those who are alive and without the primary event at the time of primary event of individual $i$.

When data are complete, that is, for each individual we observe the primary event or the competing event or the individual reaches the end of the study, the partial likelihood function for estimating the subdistribution hazard is identical to Eq (1) [9], section 3.1. The risk set $R_i$ is however augmented with individuals with competing event prior to the time of primary event of individual $i$. Indeed, in the complete CHS data, the subdistribution hazard ratio estimates are identical when fitted with traditional Cox regression with an augmentation of the time at risk for individuals with competing event. We note that in the absence of competing events there is no need to augment the risk set and the cause-specific and subdistribution hazards are identical.

With incomplete data, that is when right censoring is present, weights $w_{ij}$ are incorporated into the partial likelihood function for subdistribution hazard estimation

$$L(\beta) = \prod_{i=1}^{n} \left( \frac{\exp\{\beta^T Z_i\}}{\sum_{j \in R_i} w_{ij} \exp\{\beta^T Z_j\}} \right)^{I(D_i=k)}.$$

The weights are inverse probability of censoring weights, based on a Kaplan-Meier estimate for censoring evaluated at two different time points. Specifically, $w_{ij} = \frac{G(T_i)}{G(T_i \wedge T_j)}$ where $\wedge$ denotes minimum. The weight is one for all those who are competing event and primary event free at the time of primary event of individual $i$ and $w_{ij} = \frac{G(T_i)}{G(T_j)}$ for those with competing event before the time of primary event of individual $i$. These weights are often mistaken for competing event weights, but they do not address the probability of having a primary event for those with competing events. Rather, these weights only re-scale the part of population included in the risk set for competing events to mirror the censoring due to loss of follow-up. Primary events are not added among those with competing events, and time at risk is not shortened in those with competing events.

In the subdistribution risk set, those who have not experienced any type of event are being combined with those who experienced the competing event first. The latter subset is rather different from the former, and often so are their risk factors. We can explain the differences in the methodologies' estimates we saw with the CHS data by exploring the risk factors of the group of participants who experienced death as a competing risk. In Fig 3, we have added the

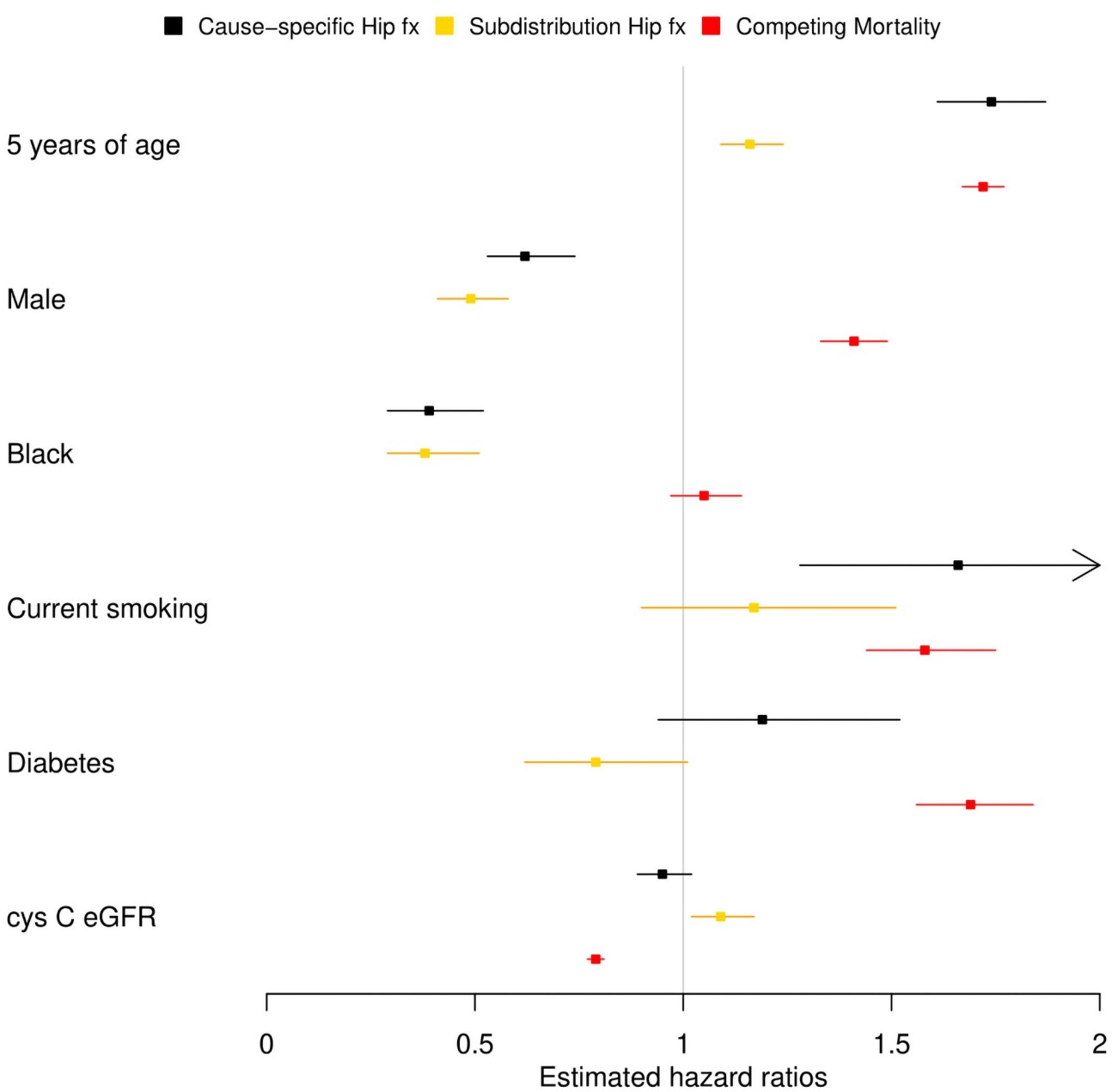

**Fig 3. Hazard ratios estimates.** Contrasting the hazard ratio estimates for incident hip fracture between the cause-specific approach and the FG subdistribution approach. Mortality cause-specific hazard ratio estimate foretells the mutual position of the hazard ratios for incident hip fracture.

cause-specific mortality estimates. Competing mortality is positively associated with age, as is incident hip fracture, and thus by augmenting the risk set for hip fracture in the FG approach with competing mortality population, we artificially lower the hazard. With gender the situation is reversed. The competing mortality association with male gender has opposite direction to the hip fracture association, and thus the FG approach suggests a stronger association with hip fracture. Lastly, when there is a lack of association of the competing mortality with black race, the FG and cause-specific approaches provide a similar hazard ratio. The authors of the clinical article further discuss the implausible associations for diabetes and cystatin C-based

eGFR observed with FG approach, in contrast to the associations observed in situations when competing death is rare [8].

To summarize, when the associations between the primary event and a risk factor and the competing event and a risk factor are concordant, then the primary event FG HR will be closer to null than the cause specific HR, and in some situations can even become discordant with the cause specific HR. In the rare situations when the associations between the primary event and a risk factor and the competing event and a risk factor are discordant, then the primary event FG HR will be further from null than the cause specific HR. We note that the ground work for a quantitative link between the FG hazard and cause-specific hazard has been recently laid out [12] and reduction factor has been introduced, representing the proportion of subjects in the FG risk set that has not yet experienced a competing event.

## Incidence rate

While hazard takes into account the relative times at which events occurred by considering the different risk sets for each person at their event time, incidence rate only measures the ratio between the number of events and the total time at risk. Because of this, incidence rate is an optimal statistic to show the conceptual difference between the cause-specific and subdistribution FG approaches.

In Table 2 we list the number of events, person-years at risk, and incidence rates along with their confidence intervals calculated using a quasi-Poisson model with offset to account for the time at risk. Competing risks are considered multi-component endpoints and it is often emphasized that all of them need to be analyzed simultaneously [13, 14]. Thus, we add the analysis of mortality in the hypothetical scenario where hip fracture is its competing event. For completeness, we also consider the composite outcome [13, 15] of hip fracture and mortality as well as that of mortality alone.

In the analysis of incident hip fracture, the CS approach and the SD approach have the same number of events (688). However, the SD approach claims almost twice as many person-years at risk than the CS approach because it keeps the 3979 individuals who die before having hip fracture in the risk set until the end of the study. Therefore, the incidence rate, reflecting both the number of events and the person-years at risk, is about half in the SD estimate as compared to the CS estimate.

Similarly, for mortality, the CS and SD approaches use the same number of deaths (3979), but the number of person years at risk is larger in the SD approach owing to those who had a hip fracture, resulting in a smaller incidence rate. In this case, SD adds fewer years to the time at risk compared to the analysis of hip fracture, leading to more similar incidence rate estimates between the SD and CS approaches. Fig 4 further shows the boxplots of years at risk across the approaches. The FG approach for hip fracture inflates the time at risk, with about

**Table 2. Summary of CS and SD approaches.**

| Outcome, approach n = 5265 | Event count | Person-years at risk | Incidence rate (IR) | 95% CI of IR |
|---|---|---|---|---|
| Hip fracture, CS | 688 | 59776 | 11.5 | (10.2, 13.0) |
| Hip fracture, SD | 688 | 107238 | 6.4 | (5.4, 7.6) |
| Mortality, CS | 3979 | 59776 | 66.6 | (63.4, 69.8) |
| Mortality, SD | 3979 | 68369 | 58.2 | (55.2, 61.3) |
| Composite event | 4667 | 59776 | 78.1 | (74.8, 81.5) |
| Mortality | 4580 | 62207 | 73.6 | (70.6, 76.8) |

Incidence rate is per 1000 person-years of follow-up, computed with Poisson model with offset for time at risk.

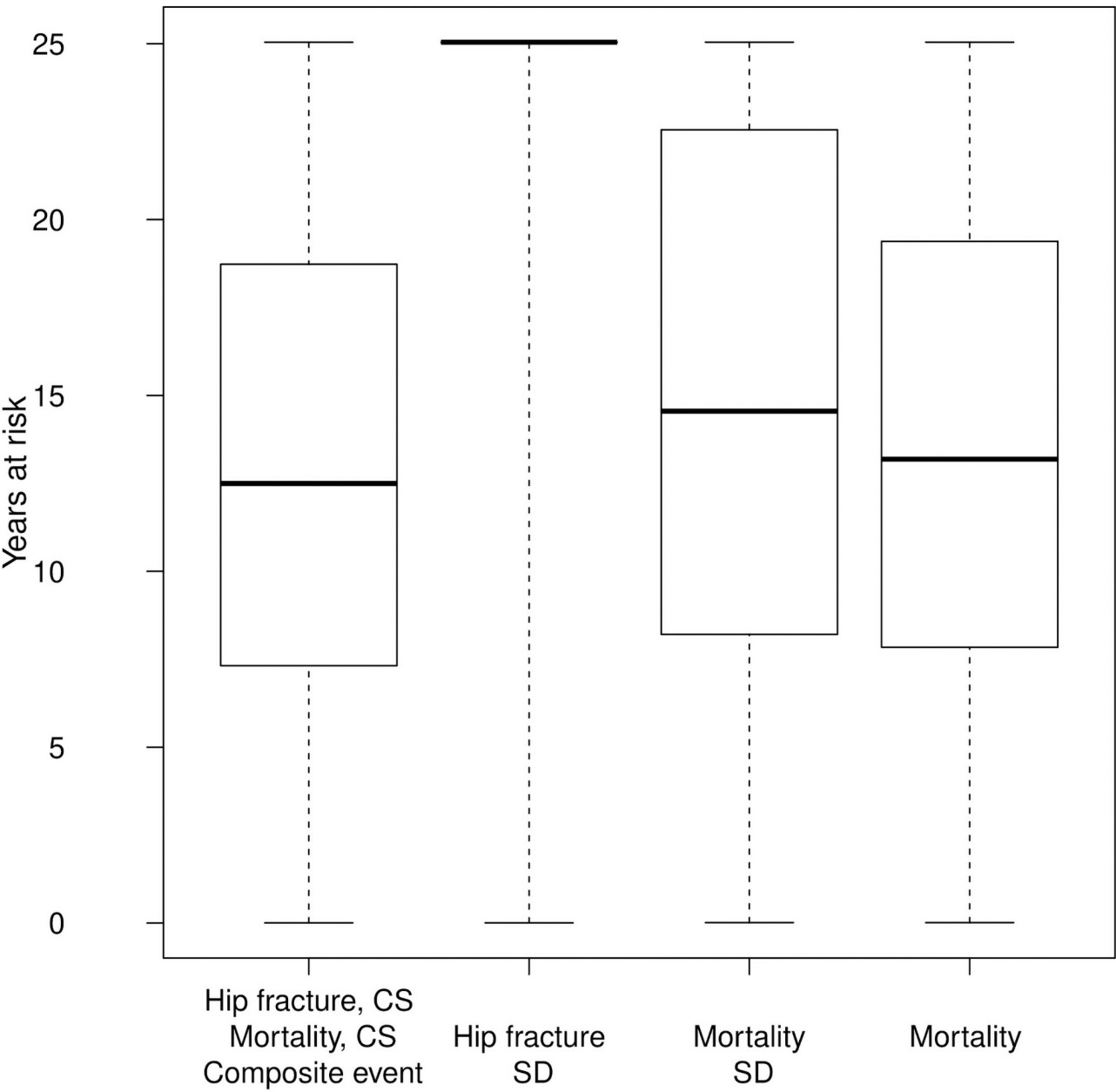

**Fig 4. Boxplots of years at risk for incident hip fracture and mortality.** Various approaches.

87% of individuals reaching the end of the study under this model (alive or dead), while only 10% of study participants were hip fracture free and alive at the end of the study.

Note that the model of composite outcome has the most events, counting both hip fractures and deaths in those without a hip fracture. The actual mortality model has a larger number of deaths than both the CS and the SD mortality approach because it includes deaths after incident hip fractures.

We can see that in incidence rate, similarly to hazard, the SD approach artificially increases the time at risk by keeping individuals who have already had a competing event as "at risk" until the end of the study. While CS estimates the risks of having hip fracture among living

**Table 3. Incidence rate ratios and hazard ratios estimates for incident hip fracture, mortality and composite event in model M1.**

| Outcome, approach | 5 years of age | | Male | | Black | |
|---|---|---|---|---|---|---|
| | **IRR** | **95% CI** | **IRR** | **95% CI** | **IRR** | **95% CI** |
| Hip fracture, CS | 1.49 | (1.35, 1.64) | 0.59 | (0.46, 0.75) | 0.4 | (0.26, 0.6) |
| Hip fracture, SD | 1.16 | (1.01, 1.33) | 0.48 | (0.34, 0.69) | 0.38 | (0.21, 0.69) |
| Mortality, CS | 1.41 | (1.35, 1.47) | 1.36 | (1.23, 1.49) | 1.11 | (0.98, 1.26) |
| Mortality, SD | 1.3 | (1.24, 1.36) | 1.45 | (1.31, 1.61) | 1.21 | (1.06, 1.39) |
| Composite Event | 1.42 | (1.37, 1.48) | 1.21 | (1.11, 1.32) | 0.99 | (0.88, 1.11) |
| Mortality | 1.43 | (1.37, 1.48) | 1.28 | (1.17, 1.4) | 1.04 | (0.92, 1.16) |
| | **HR** | **95%CI** | **HR** | **95%CI** | **HR** | **95%CI** |
| Hip fracture, CS | 1.74 | (1.61, 1.87) | 0.62 | (0.53, 0.74) | 0.39 | (0.29, 0.52) |
| Hip fracture, SD | 1.16 | (1.09, 1.24) | 0.49 | (0.41, 0.58) | 0.38 | (0.29, 0.51) |
| Mortality, CS | 1.7 | (1.65, 1.75) | 1.47 | (1.38, 1.56) | 1.12 | (1.03, 1.21) |
| Mortality, SD | 1.36 | (1.33, 1.4) | 1.54 | (1.44, 1.63) | 1.25 | (1.15, 1.36) |
| Composite Event | 1.7 | (1.66, 1.75) | 1.31 | (1.23, 1.38) | 0.99 | (0.92, 1.07) |
| Mortality | 1.72 | (1.67, 1.77) | 1.41 | (1.33, 1.49) | 1.05 | (0.97, 1.14) |

populations, the SD approach estimates the risks of having a hip fracture among both those alive and those who have already died.

From now on, we consider only model M1 with the three demographic risk factors of age, gender, and race. The incidence rate ratios are in the upper part of (Table 3), with hazard ratios for comparison in the lower part.

We can see similar relationships between incidence rate ratios calculated using the SD and CS approaches as have been discussed for hazard ratios.

## Cumulative incidence

It is often claimed that the FG hazard approach estimates hazard ratios that are directly linked to the effect of a risk factor on the cumulative incidence of events. Fig 5 shows the cumulative incidence function (CIF) estimates for the six scenarios.

The dark red curve representing the hip fracture CIF estimate in the CS approach computes the CIF among individuals alive at the time. We see that this curve is higher than the red dashed curve, the hip fracture CIF estimate using the SD approach, where individuals after death are maintained in the sample as hip fracture-free. Thus, while the CIF under the SD approach may be a useful statistic for public health decisions, or prediction, its use for estimating the associations between an individual's outcome and a risk factor is limited.

Similar observations apply to the blue curves when we estimate the CIF of mortality with incident hip fracture as competing event. The black line is the true mortality CIF estimate. We note that its upper boundary is the composite event CIF (green curve) and its lower boundary is the mortality CIF of the SD approach. The same can be said for hip fracture.

The incident hip fracture and mortality CIF in the subdistribution approach sum up to the composite event CIF. For heterogeneous events such as any non-fatal event and mortality, we find this to be of no actual advantage.

When there is no loss of follow-up, we can directly model the CI under the subdistribution approach. Cumulative hazard is naturally related to the cumulative incidence function through the complimentary −log link [9]. Thus, we model CI at the end of the study with $-\log(1 - CI)$ = $\exp\{\beta^T Z\}$. Table 4 shows these estimates, often called cumulative hazard ratio (CHR) estimates, for model M1 with the demographic risk factors for age, gender, and race. These

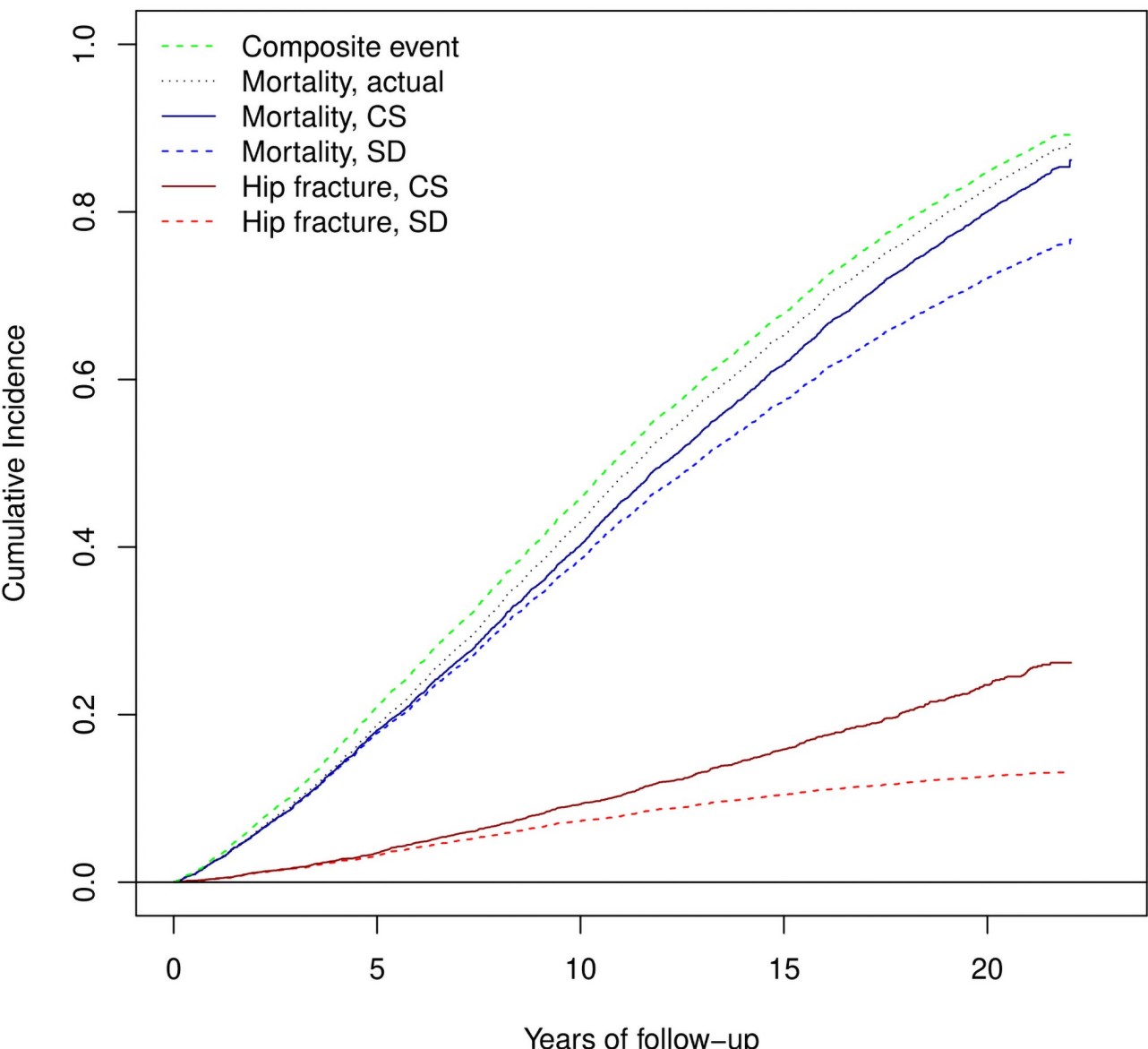

**Fig 5. Estimates of cumulative incidence.** Cumulative incidence functions estimates for incident hip fracture, mortality and composite event.

estimates are indeed similar to the FG hazard ratios, showed at the top part of the table. However, there is not a direct link between the FG hazard ratios and relative changes in CI when modeled directly [2]. To demonstrate that we model the cumulative incidence relative risk (RR) with quasi-Poisson approach, CI = $\exp\{\beta^T Z\}$. Additionally, we model the cumulative incidence odds ratios (OR), $\frac{\text{CI}}{1-\text{CI}} = \exp\{\beta^T Z\}$, both at the lower part of Table 4. We see large differences between the hazard ratios and the relative risks and odds ratios, especially in the associations of risk factors with mortality. This is not surprising as one is a hazard and the other two are cumulative risks.

We note that similarly to the link between cumulative incidence and hazard under the subdistribution approach, there is an identical link between the cumulative incidence and hazard under the cause-specific approach in the absence of competing events. In both situations, the

**Table 4. The comparison of hazard ratios estimates, cumulative hazard ratios estimates, relative risk estimates, and odds ratios estimates for cumulative incidence for incident hip fracture and mortality in the competing risk setting under the subdistribution approach.**

| Statistic | 5 years of age | | Male | | Black | |
|---|---|---|---|---|---|---|
| **Outcome** | | | | | | |
| log Hazard | **HR** | **95% CI** | **HR** | **95% CI** | **HR** | **95% CI** |
| Hip fracture | 1.16 | (1.09, 1.24) | 0.49 | (0.41, 0.58) | 0.38 | (0.29, 0.51) |
| Mortality | 1.36 | (1.33, 1.4) | 1.54 | (1.44, 1.63) | 1.25 | (1.15, 1.36) |
| $\log(-\log(1-CI))$ | **CHR** | **95% CI** | **CHR** | **95% CI** | **CHR** | **95% CI** |
| Hip fracture | 1.15 | (1.07, 1.23) | 0.49 | (0.41, 0.58) | 0.38 | (0.29, 0.51) |
| Mortality | 1.3 | (1.26, 1.34) | 1.45 | (1.35, 1.55) | 1.26 | (1.14, 1.38) |
| logCI | **RR** | **95% CI** | **RR** | **95% CI** | **RR** | **95% CI** |
| Hip fracture | 1.13 | (1.06, 1.2) | 0.51 | (0.44, 0.6) | 0.4 | (0.31, 0.52) |
| Mortality | 1.08 | (1.07, 1.09) | 1.17 | (1.14, 1.21) | 1.1 | (1.06, 1.14) |
| logit CI | **OR** | **95% CI** | **OR** | **95% CI** | **OR** | **95% CI** |
| Hip fracture | 1.14 | (1.07, 1.22) | 0.46 | (0.39, 0.56) | 0.36 | (0.27, 0.48) |
| Mortality | 1.47 | (1.37, 1.57) | 2.04 | (1.78, 2.34) | 1.55 | (1.29, 1.86) |

link is however indirect; a hazard ratio estimate is only linked to the complimentary −log link of cumulative incidence, and cannot be interpreted as effects of risk factors on the cumulative incidence.

## Discussion

The FG subdistribution methodology keeps individuals with competing events in the study, while forever curing them of the primary event. It is thus not suitable when estimating the association between a new biomarker, or drug, and a non-fatal event [3, 8, 13, 16]. In such scenarios, the population of interest is those without the competing event, which is modeled by the cause specific approach. Indeed, application of FG methods to these scenarios can often lead to results that conflict with well-established associations. FG approach doesn't address missing data problems related to informative censoring due to competing risk, but simply changes the population studied. According to its creators, the approach is "better suited for estimating a patient's clinical prognoses" [3], a scientific question asked less often. A complication for prognostic applications of the FG methodology is that prognostic models are often developed from data collected decades ago, and as such the effects of death being protective of the primary outcome are overestimated because of increasing life expectancy. Individuals who died in the study and were thus cured of the primary event may currently stay alive and be at risk.

We note that Lunn McNeil approach [17] to competing risk in modeling hazard is sometimes used as an alternative the CS and FG approaches. It is based on the cause-specific approach, and estimates hazard ratios simultaneously for all types of events by augmenting the data, treating other types of events as censoring. It allows for comparison of magnitude of hazard ratios across the types of the events. A drawback is that it assumes independent risk, that is, non-informative censoring due to the other competing events. This assumption of independent risk often limits the use of this methodology. For example, in the CHS study, mortality hazard is likely not independent of incident hip fracture hazard, so any estimates obtained from this approach may be biased and the comparisons between hazard ratios for different types of events may be invalid. Further research is needed into bias of this approach under competing risks and how to modify this approach to provide a valid estimate comparison.

Sometimes, FG methodologies are used because researchers believe that the proportional hazard assumption of Cox regression prevents the CS approach from being valid. However, both FG subdistribution hazard and the cause specific approaches are modeled using Cox regression. If the proportional hazard assumption is violated, the estimator is still well defined: it is the "average hazard" over time [18, 19]. If an average hazard is not of interest, there are two simple ways to address the problem. One approach is to accommodate non-proportional hazards by including interactions between the risk factor and time in the Cox regression model as time-dependent predictors. A second approach is to divide the data into strata based on the value of the risk factor, with each stratum permitted to have a different baseline hazard function.

We now give the correct interpretations for the two methodologies. Cause-specific Cox regression can be fitted in R by using package survival, with coxph function. It estimates the hazard of the primary event in the population free of the primary event and the competing event. To estimate hazard ratios, we contrast a person with a primary event with those without a primary or competing event by the time. In the context of the CHS data, we contrast an alive individual who sustained hip fracture to those alive who have not sustained hip fracture by that time.

FG subdistribution hazard can be estimated in R by using package cmprsk with crr function. It estimates the hazard of the primary events in the population free of the primary event. This population includes those that sustained the competing event. To estimate hazard ratios, we contrast a person with a primary event with those without primary event by that time. We contrast an alive individual who sustained hip fracture to those alive who have not sustained hip fracture by that time augmented with individuals why had died by that time without having a hip fracture. The estimated hazard ratios cannot be interpreted as estimates of the effect of a risk factor on the cumulative incidence of events.

In conclusion, we find that considering the population of interest is critical to choosing the correct methodology. If the population of interest is individuals free of a competing event at a given time, the cause-specific approach should be used. This is commonly the case when death is a competing event, as the clinical interest lies with alive individuals. In such a population, death does not occur by definition, and therefore should not be considered a competing event.

If the population of interest is the entire starting population, but the competing events are non-informative about the risk of primary events, the cause-specific approach should likewise be used. If the population of interest is the entire population, the competing events are informative, and one is interested in what would have happened had the competing event not occurred, simulations should be used with a list of scenarios of the dependence structure between the primary and competing events.

## Author Contributions

**Conceptualization:** Petra Buzkova.

**Formal analysis:** Petra Buzkova.

**Methodology:** Petra Buzkova.

**Writing – original draft:** Petra Buzkova.

**Writing – review & editing:** Petra Buzkova.

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
