## [Decision Letter · Decision Letter 0]

6 May 2021

PONE-D-21-10045

Competing Risk of Mortality in Association Studies of Non-fatal Events

PLOS ONE

Dear Dr. Buzkova,

Thank you for submitting your manuscript to PLOS ONE. After careful consideration, we feel that it has merit but does not fully meet PLOS ONE’s publication criteria as it currently stands. Therefore, we invite you to submit a revised version of the manuscript that addresses the points raised during the review process.

Reviewer 1 has asked that you include some additional information about the statistical modeling to make your analyses easier to understand by non-specialists.

We look forward to receiving your revised manuscript.

Kind regards,

Robert Daniel Blank, MD, PhD

Academic Editor

PLOS ONE

Journal Requirements:

In your Data Availability statement, you have not specified where the minimal data set underlying the results described in your manuscript can be found. PLOS defines a study's minimal data set as the underlying data used to reach the conclusions drawn in the manuscript and any additional data required to replicate the reported study findings in their entirety. All PLOS journals require that the minimal data set be made fully available. For more information about our data policy, please see http://journals.plos.org/plosone/s/data-availability.

Thank you for stating the following in the Acknowledgments Section of your manuscript:

This research was supported by contracts HHSN268201200036C, HHSN268200800007C,

HHSN268201800001C, N01HC55222, N01HC85079, N01HC85080, N01HC85081,

N01HC85082, N01HC85083, N01HC85086, and grants U01HL080295 and U01HL130114

from the National Heart, Lung, and Blood Institute (NHLBI), with additional

contribution from the National Institute of Neurological Disorders and Stroke (NINDS).

Additional support was provided by R01AG023629 from the National Institute on

Aging (NIA). A full list of principal CHS investigators and institutions can be found at

CHS-NHLBI.org. The content is solely the responsibility of the authors and does not

necessarily represent the official views of the National Institutes of Health.

Reviewers' comments:

Reviewer's Responses to Questions

**Comments to the Author**

1. Is the manuscript technically sound, and do the data support the conclusions?

Reviewer #1: Yes

Reviewer #2: Yes

2. Has the statistical analysis been performed appropriately and rigorously? 

Reviewer #1: Yes

Reviewer #2: I Don't Know

3. Have the authors made all data underlying the findings in their manuscript fully available?

Reviewer #1: Yes

Reviewer #2: Yes

4. Is the manuscript presented in an intelligible fashion and written in standard English?

Reviewer #1: Yes

Reviewer #2: Yes

5. Review Comments to the Author

Reviewer #1: This paper compares 2 commonly used time-to-event methods: cause specific and subdistribution hazards models applied to studies of non-fatal events. The topic is important, and the manuscript covers well the main points of interest.

I have several comments:

1. In the Introduction, the author stated that one of the main current issue is related to the misuse of terminology when FG subdistribution methodology is used. I indeed agree with the author on this. However, I could not find in the manuscript, any suggestion from the author on an appropriate use of terminology. It would be great, if such a paragraph with appropriate reporting and terminology can be included in the manuscript.

2. A diagram depicting the differences in allocation of person risks between the 2 methodologies would be helpful, particularly for the readers without a strong statistical background. The section on incidence rates is particularly difficult to follow. A clear definition including allocation of events, person-years and deaths should accompany that section.

3. It will also be useful to add a section on the availability and syntaxes for FG methodology in common statistical softwares such as SAS, Stata, R. This section could also include the appropriate use of terminology for reporting in future publication

4. In the discussion, a point is made that FG subdistribution hazard model is generally recommended for clinical prognostic models. However, there can be some limitations. For example, the author may consider commenting on the impact of increasing population’s life expectancy on the validity of prognostic models using competing risk of mortality, particularly that these prognostic models are usually developed in study cohorts collected 2 decades prior to their intended use, when mortality risk was much higher than in the current context.

Reviewer #2: I don't have specific suggestions for the authors.

The manuscript looks like a PFD of a paper. Has this been published before?

Would suggest submitting revision double spaced with line numbers.

6. PLOS authors have the option to publish the peer review history of their article (what does this mean?). If published, this will include your full peer review and any attached files.

Reviewer #1: No

Reviewer #2: No

---

## [Author Response · Author response to Decision Letter 0]

25 Jun 2021

We attached a file with response to reviewers.

---

## [Decision Letter · Decision Letter 1]

14 Jul 2021

Competing Risk of Mortality in Association Studies of Non-fatal Events

PONE-D-21-10045R1

Dear Dr. Buzkova,

We’re pleased to inform you that your manuscript has been judged scientifically suitable for publication and will be formally accepted for publication once it meets all outstanding technical requirements.

Kind regards,

Robert Daniel Blank, MD, PhD

Academic Editor

PLOS ONE

Additional Editor Comments (optional):

Reviewers' comments:

Reviewer's Responses to Questions

**Comments to the Author**

1. If the authors have adequately addressed your comments raised in a previous round of review and you feel that this manuscript is now acceptable for publication, you may indicate that here to bypass the “Comments to the Author” section, enter your conflict of interest statement in the “Confidential to Editor” section, and submit your "Accept" recommendation.

Reviewer #1: All comments have been addressed

2. Is the manuscript technically sound, and do the data support the conclusions?

Reviewer #1: Yes

3. Has the statistical analysis been performed appropriately and rigorously? 

Reviewer #1: Yes

4. Have the authors made all data underlying the findings in their manuscript fully available?

Reviewer #1: Yes

5. Is the manuscript presented in an intelligible fashion and written in standard English?

Reviewer #1: Yes

6. Review Comments to the Author

Reviewer #1: I thank the author for addressing all my comments. I think that the manuscript is much clearer now. I have no further comments.

7. PLOS authors have the option to publish the peer review history of their article (what does this mean?). If published, this will include your full peer review and any attached files.

Reviewer #1: No

---

## [Editor Report · Acceptance letter]

2 Aug 2021

PONE-D-21-10045R1 

Competing risk of mortality in association studies of non-fatal events 

Dear Dr. Buzkova:

I'm pleased to inform you that your manuscript has been deemed suitable for publication in PLOS ONE. Congratulations! Your manuscript is now with our production department. 

Kind regards, 

on behalf of

Professor Robert Daniel Blank 

Academic Editor

PLOS ONE